# An assessment of turnaround times of infant Deoxyribonucleic acid–Polymerase Chain Reaction testing and the associated factors in Western Kenya: A mixed methods study

**Maxwell Philip Omondi** *

School of Public Health and Community Development, Maseno University, Maseno, Kenya

* maxwellomondi@gmail.com

## Abstract

### Introduction

The HIV/AIDS continues being a significant global public health priority in the 21[st] century with social and economic consequences Mother-to-child transmission (MTCT) occurs when an HIV-infected woman passes the virus to her infant and about 90% of these MTCT infections occurs in Africa where children and infants are still dying of HIV. Early definitive diagnosis using Deoxyribonucleic acid reaction of HIV infection in infants is critical to ensuring that HIV-infected infants receive appropriate and timely care and treatment to reduce HIV related morbidity and mortality.

### Objective

To assess the Infant Deoxyribonucleic acid–Polymerase Chain Reaction (DNA-PCR) Turnaround Time (TAT) of dry blood spots and associated factors in Vihiga, Bungoma, Kakamega and Busia counties, in Kenya.

### Method

A mixed methods study using a) retrospectively collected data from Ministry of Health Laboratory registers, Early Infant Diagnosis (EID) database from 28 health facilities and b) 9 key informant interviews with laboratory in-charges were conducted. A total of 2,879 HIV exposed babies' data were abstracted from January 2012 to June 2013.

### Results

The mean TAT from specimen collection and results received back at the facilities was 46.90 days, Vihiga county having the shortest mean duration at 33.7days and Kakamega county having the longest duration at 51.7days ($p = 0.001$). In addition, the mean transport time from specimen collection and receipt at Alupe Kenya Medical Research Institute (KEMRI) reference Laboratory was 16.50 days. Vihiga County had the shortest transport time at 13.01 days while Busia had the longest at 18.99 days ($p = 0.001$). Longer TAT was

**Data Availability Statement:** All relevant data are within the manuscript and its Supporting Information files.

**Funding:** The author(s) received no specific funding for this work.

**Competing interests:** The authors have declared that no competing interests exist.

due to the batching of specimens at the peripheral health facilities and hubbing to the nearest referral hospitals.

## Conclusion

The TAT for DNA-PCR specimen was 46.90 days with Vihiga County having the shortest TAT due to lack of specimen batching and hubbing.

## Recommendation

Discourage specimen batching/hubbing and support point-of-care early infant diagnosis (EID) tests.

## Background

The HIV/AIDS continues being a significant global public health priority in the 21$^{st}$ century with social and economic consequences. In 2022, there were approximately 39million people living with HIV, of which 1.5million were children and 53% of all people living with HIV were women and girls [1]. In sub-Saharan Africa, women and girls (of all ages) accounted for 63% of all new HIV infections in 2022 [2].

Fewer new HIV infections in women and high treatment coverage among women living with HIV have led to a steep drop in the annual number of new vertical infections in children, which fell by 58% between 2010 and 2022. About 93% of pregnant or breastfeeding women living with HIV were receiving antiretroviral therapy in 2022, up from 48% in 2010 in eastern and southern Africa [2].The means for eliminating new HIV infections in children exist, but gaps in services to prevent vertical transmission of HIV still leave hundreds of thousands of children at high risk of acquiring HIV each year [2].

Children are still much less likely than adults to receive antiretroviral therapy. Coverage among children living with HIV was 57% in 2022, compared with 77% among adults—and that gap is widening. Approximately 660 000 children living with HIV were not receiving antiretroviral therapy in 2022. As a result, children accounted for 13% of AIDS-related deaths in 2022, even though they comprise only about 4% of people living with HIV [2]. The 2018 Kenya Population-Based HIV Impact Assessment estimated the burden and prevalence of paediatric HIV infection at 138,900 in Kenya [3].

Wider adoption of point-of-care early infant diagnosis will help close that gap by allowing test results to be produced at or near the site of patient care. Point of Care (POC) early infant diagnosis (EID) ensures that infants are tested onsite, and that their caregivers receive their test results quickly–often on the same day or during the same clinic visit. POC testing, when integrated into a national laboratory network, is a useful complement to existing centralized laboratory testing, particularly for populations such as infants needing their test results returned quickly [4]. The point-of-care has the potential of drastically reducing the TAT to an average to 2 days, leading to 100% HIV-infected infants being initiated on ART [5]. WHO recommends that all infants exposed to HIV receive a virological HIV test within two months of birth. In a randomized study in Mozambique and the United Republic of Tanzania, AIDS-related deaths among infants in their first six months of life were reduced by 73% when point-of-care diagnosis was provided and rapid linkage to antiretroviral therapy was achieved [6]. Early infant diagnosis coverage has risen in eastern and southern Africa to 83% [2].

Mother-to-child transmission (MTCT) occurs when an HIV-infected woman passes the virus to her baby. This can occur before, during, or after delivery and have high mortality, with 50% dying before one year of age, and 20% of these early deaths occurring between the first and third months of life [7].

Strong evidence indicates that combination antiretroviral therapy (ART) should be initiated in HIV-infected infants early in the first year of life to reduce morbidity and mortality [8]. In 2008, there was a paradigm shift away from initiating ART in children based on clinical and/or immunologic status to starting ART in all children <1 year of age as soon as diagnosed regardless of clinical and immunological status [9]. Support for this shift was largely due to results from the South African Children with HIV Early Antiretroviral Therapy (CHER) randomized trial, which showed a reduction in mortality by 76% among infants randomized to immediate ART compared to deferred ART [10, 11].

While Early Infant Diagnosis (EID) and treatment reduces mortality by up to 76%, universal testing of HIV-exposed infants (HEI) born to HIV-positive mothers has not yet been achieved [10, 12]. Standard HIV antibody testing—as is done with adults and older children—cannot identify infected infants in their first year of life, as it also detects maternal HIV antibodies that are transferred to the baby during pregnancy and can persist up to 18 months of age. More demanding testing methods namely Deoxyribonucleic acid—Polymerase Chain Reaction (DNA-PCR) that rely on detecting HIV virus, or virological tests are required for diagnosing infants before 18 months of age [13, 14]. For this reason, PCR—based viral nucleic acid tests for dry blood spots remain standard laboratory diagnosis for HIV early infant diagnosis [15]. At present, point of care viral DNA or RNA tests for dry blood spot are now available that require minimal technical and infrastructural capacity to perform DNA-PCR.

The demonstration that HIV RNA and DNA can be detected in dried blood spots (DBS) revolutionized newborn HIV testing in resource-poor areas in which laboratory facilities are limited and ill-equipped [16]. Unlike serum samples that must either be tested within hours of collection or frozen for transport, DBS can be stored in warm, humid climates and later transported to reference laboratories for testing while still yielding accurate results [17]. This led the World Health Organization (WHO), United Nations Children's Fund (UNICEF), and the Centers for Disease Control and Prevention (CDC) to endorse DBS testing as the single screening tool for all children less than 18 months born to mothers with HIV infection or unknown HIV status despite the limited availability of the laboratory services especially in developing countries such as sub-Saharan Africa [13, 18, 19].

Early Infant Diagnosis (EID) coverage is still unacceptably low, despite scale-up of laboratory capacity for virological testing and implementation of larger dried blood spot testing networks. Some of the challenges of EID include insufficient infrastructure, training of staff, quality assurance in point-of-care testing, frequent stock-outs of laboratory reagents, lack of access to EID for children born to women living with HIV and operational barriers such as turnaround time for results and loss to follow up [20–22].

Since 1989 it has been shown that PCR testing can be conducted on either whole blood or dried blood spots (DBS) from infants. The use of DBS requires only a few drops of blood from an infant and the blood is dripped onto filter paper. Once specimens are collected, they can be easily stored and transported in a sealed bag or envelope without cold-chain systems to centralized testing locations with PCR technology for infant HIV testing. DBS can be easily transported, is relatively inexpensive, and requires less blood from the infant. The use of DBS permits blood samples to be collected in remote locations and allows countries with a limited number of specialized laboratories to expand access to virological testing. By providing accurate and early diagnosis of infants, DBS offers promise for more timely access to lifesaving treatment and care services for infants who are infected [23].

Early Infant Diagnosis begun in Kenya in 2005, on a small scale in Nairobi and Busia, but has been expanded to the rest of the country. Currently, there are twelve (12) testing laboratories that are based in Nairobi (KEMRI HIV Research Laboratory), Kericho (KEMRI-WRP CRC lab), Kisumu (KEMRI-CDC Lab), AMPATH CARE (Eldoret), AMPATH Reference Laboratory (Eldoret), National Public Health Reference Laboratory (Nairobi), IDAP (Nairobi), Nyumbani Children Diagnostic Laboratory (Nairobi),Kenyatta National Hospital (Nairobi), Kenyatta University Teaching and Referral Hospital (Kiambu), Coast County Teaching and Referral Hospital and Busia (KEMRI-ALUPE Lab).

Point-of care (POC) testing for early infant diagnosis was introduced in Kenya in 2017 as a pilot project in Turkana and Homabay counties under the funding of Unitaid [5, 24, 25]. It was then scaled up to the rest of the country. At the end of the Unitaid funded project in 2019, the county governments were expected to take it up and absorb the running costs of POC. This has however, faced challenges with frequent stock-outs of laboratory reagents. This has effectively resulted in the four counties under study sending their DBS samples to Alupe KEMRI Reference Laboratory.

In May 2006 MOH introduced a national EID algorithm recommending PCR testing for dry blood spots in all HIV exposed infants from 6 weeks with confirmatory antibody test at 18 months. This has since been updated with 2016 edition that replaced 9-month, serology-based EID testing with virologic EID testing at 6 and 12 months of age.

In Busia, Kakamega, Vihiga and Bungoma counties in Western region of Kenya, Alupe KEMRI Reference Laboratory serves as the reference laboratory for EID of HIV-exposed children or those with unknown status. Laboratory specimens were collected at peripheral health facilities and transported through a CD4 laboratory networking program that was facilitated by Courier services funded by USAID. The specimens were delivered to Alupe KEMRI Reference Laboratory and the results dispatched back to health facilities through the same laboratory networking arrangement.

Turnaround time (TAT) is the total time between specimen collection, submission, processing and dispatch of the results for patient use. TAT is one of the most noticeable signs of laboratory service and is often used as a key performance indicator of laboratory performance. It is useful as a source of benchmarking laboratory performance and as measure for continuous quality improvement. It impacts clinical outcomes. Over 80% of laboratories receive complaints about TAT, yet there is little agreement among clinicians on what constitutes acceptable TAT though WHO recommends that caregivers receive the child's test results within 30 days [23, 25–27]. Long TAT for DNA-PCR results can lead to preventable deaths, particularly given that the approximately 30% mortality of perinatally-infected infants during the first six months of life [7] and the advanced stage of HIV disease that affects approximately half of HIV-infected infants who do not start antiretroviral therapy (ART) before 12 weeks of age [28]. In some countries, site level batching has significantly increased TAT and low rates of result return within 30 days. In cases where there is an additional step in the transport network between the site and the central lab (e.g., a hub site) TAT for EID result increased greatly. Even when results are returned within 30 days, many patients never receive their results, underscoring other challenges with counseling, patient follow up and data systems [29]. One challenge is that the type of laboratory which can support sophisticated PCR equipment for dry blood spots is often only available at a referral point or center of excellence, although blood samples can be taken from more remote locations and brought into the central laboratory.

Turnaround time for the DNA-PCR based test and the associated factors has not been assessed in Vihiga, Kakamega, Bungoma and Busia Counties in Kenya. This study therefore sought to assess the TAT in the four counties and the associated factors. This would provide local data that would go a long way in contributing towards addressing the quality of pediatric

HIV management in the study area through timely intervention. The results would also guide the policy formulation on EID scale up at national and sub-national levels with regards to timely laboratory support towards pediatric HIV care in Western region and Kenya in general.

## Materials and methods

### Study design

This was a mixed methods study.

### Study population

HIV-exposed infants and Laboratory in-charges in Vihiga, Kakamega, Bungoma and Busia counties.

### Study period

January 2012 to June 2013.

### Recruitment period

1st September 2014 to 30th November 2014.

### Sample size

a) 2,789 HIV-exposed infants' records were abstracted from the 28 health facilities using retrospectively collected data from the MOH EID database and Laboratory registers and b) Nine [9] Key Informant Interviews with laboratory in-charges at the health facilities, sub-county and county health management teams across the four counties.

### Sampling technique

Multi-stage sampling technique was used to sample the 28 health facilities.

### Data collection procedures

a. The abstracted data MOH EID database and Laboratory registers were transferred to survey tracking form which was the primary data collection tool. This was done by the trained research assistant. This was then entered into an excel spread sheet and later exported to SPSS version 27.0 for data analysis.

b. Key Informant Guide was used to interview the nine (9) KIIs to determine the factors associated with TAT in the four counties. The KIIs were done at the work stations of the key informants where visual and auditory privacy was assured

### Inclusion criteria

a. Health facilities providing PMTCT and Early Infant Diagnosis as per the MOH/NASCOP protocol and guidelines in Vihiga, Kakamega, Bungoma and Busia Counties (formerly Western Province);

b. County Health Directors and Health facilities who were willing to provide administrative approval to participate in the study;

c. Health facilities that started providing early infant diagnosis services from January 2012;

d.  Mother-infant pairs that were enrolled in the sampled health facilities between January 2012 to June 2013;

e.  Key Informants who voluntarily consent to participate in the study.

## Consenting procedures

A written Informed consent was obtained from County Directors of Health from Vihiga, Kakamega, Bungoma and Busia counties as well as KEMRI Alupe Reference Laboratory Office in Busia to allow access to MOH EID database and Laboratory registers for the health facilities under the study area.

a.  Written informed consents were also obtained from the nine (9) Key Informants before the key informant interviews (S2 Text KII Keys).

## Data management and analysis plan

**a) Quantitative data.**  Quantitative data extracted from the MOH EID database (S1 File PCR database) and Laboratory HEI registers were transferred to survey tracking form developed for the study (S3 Text Data abstraction survey tool). These were then transferred to the excel sheet and exported to SPPS (Statistical Package for Social Sciences version 27.0) for analysis. Descriptive statistics were done using frequency distribution and measures of central tendencies while inferential statistics were done using the Analysis of Variance (ANOVA) technique.

**b) Qualitative data.**  A qualitative guide (S1 Text KII Guide) was used to collect qualitative data. The qualitative data were obtained through verbatim written recordings by the Principal Investigator (PI) and were then subjected to thematic analysis using NVIVO 12 Pro. The research question was reviewed and imported into NVivo for ease of reference. Summary memos [for key points] were written after going through a few transcripts and the word document (generated from the data abstractions). Summary memos for the transcripts were reviewed and coding strategy developed by noting down key issues coming from the interviews on the transcripts and word document. Seeing how these key issues relate to the research questions, a coding framework was developed [Generation of Nodes/Codes]. Transcripts and word document were imported into NVIVO for coding process. It entailed going through the texts on the transcripts and word document, dragging and dropping on the correct nodes. Some codes were modified and other new ones added along the way. Reviewing what was coded to see if the texts were on the correct nodes. Checking the frequencies and arranging the codes. Themes were identified from arranged codes and those related to the objective of the study were analysed and noted down. Samples of quotations were noted down under the analysed themes.

## Ethical considerations

This protocol was reviewed and approved by the Kenyatta National Hospital/University of Nairobi Ethics and Research Committee (P66/11/2012). Administrative approval was granted by the County Health Directors and the health facility-in charges of Kakamega, Vihiga, Busia and Bingoma counties.

**Table 1. Sociodemographic characteristics of the sample population (n = 2879).**

| Variable | Categories | Frequency (n, %) |
| --- | --- | --- |
| County | Bungoma | 608 (21.1%) |
| | Busia | 465 (16.2%) |
| | Kakamega | 1239 (43.0%) |
| | Vihiga | 567 (19.7%) |
| Sex | Female | 1400 (51.9%) |
| | Male | 1299 (48.1%) |
| | Not stated | 180 |

Not stated refers to missing records in the EID registers and were excluded from the analysis.

## Results

### A) Data abstraction results

**Socio-demographic characteristics.** Overall, 2879 data were extracted from the EID database and MOH registers. Majority of the infants receiving DNA-PCR for EID were from the Kakamega County with the least from Busia County. Females represented 51.9% (n = 1400), while males were 48.1% (n = 1299) and 6.3% (n = 180) sex were not stated in the Ministry of Health Early Infant Diagnosis register (Table 1).

The modal age for infants undergoing early infant diagnosis was 1.50 months (Fig 1).

The median age of infants was 2.0 (Inter Quartile Range: 1.5–6.0) months.

### Infant Deoxyribonucleic acid -Polymerase Chain Reaction turnaround time

Global TAT is the time from from sample collection to return of results to providers and caregivers back at the health facility. The Global TAT was 46.90 days with the dispatch time (25.60 days) consuming more than half of that time (Fig 2). The transport time from specimen collection to arrival at Alupe KEMRI Reference Laboratory was 16.46 days (Fig 2).

The mean transport time duration from specimen collection at the health facilities and receiving specimens at Alupe KEMRI Reference Laboratory was 16.46 (95% CI: 15.92–16.99) days with variations across the four counties. Vihiga County had the least mean duration at 13.01 (95% CI: 12.14–13.89) days while Busia County despite being co-located with Alupe

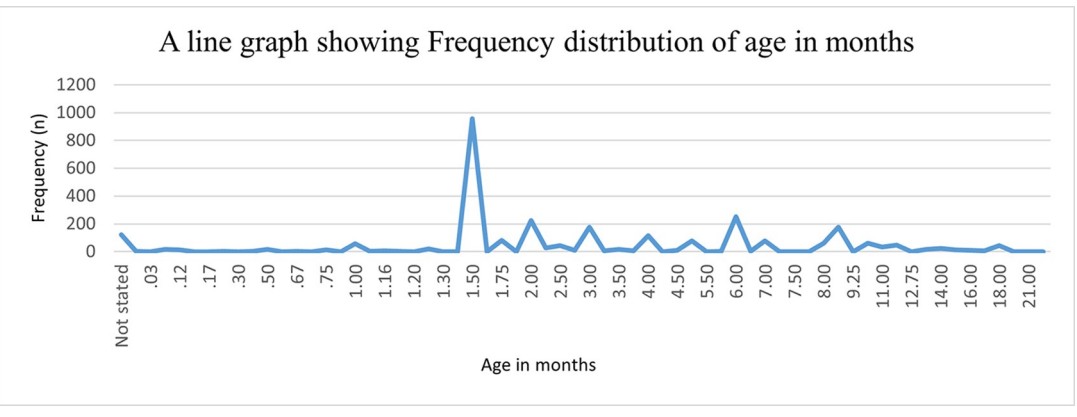

**Fig 1. Frequency distribution of age in months for infants undergoing early infant diagnosis.**

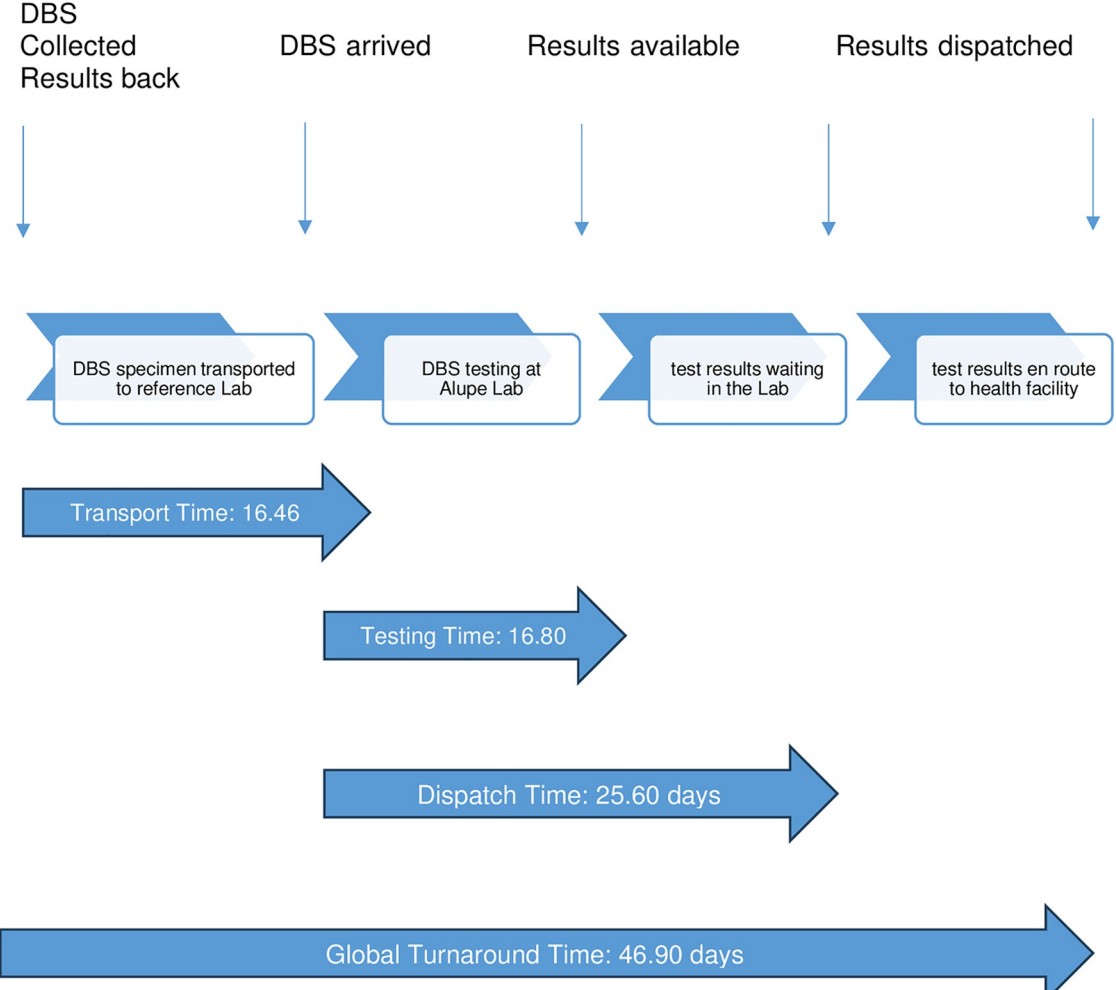

**Fig 2. Schematic representation of turnaround time.** The turnaround times are means number of days.

KEMRI Reference Laboratory had the longest duration at 18.99 (95% CI: 17.68–20.30) days (p = 0.001) (Table 2).

The mean testing time duration from receiving the specimens from the health facilities and testing the specimens was 16.80 (95% CI: 16.23–17.38) days. Though there were county level variations, these were not statistically significant (p = 0.085) (Table 2).

The mean dispatch time duration from receiving specimens at Alupe KEMRI Reference Laboratory to dispatching the results to the health facilities was 25.60 (95% CI: 24.72–26.47) days. Vihiga County had the least time at 22.89 (95% CI: 21.32–24.46) days while Kakamega County had 27.28 (95% CI: 25.70–28.86) days (p = 0.002) (Table 2).

The global means TAT from specimen collection at the health facilities and results received back at the health facilities was 46.90 (95% CI: 43.37–50.43) days with Vihiga having the least means duration at 33.67 (95% CI: 30.27–37.06) days and Kakamega having the longest means duration at 51.70 (95% CI: 33.81–42.49) days (p = 0.001) (Table 2).

**Table 2. Infant DNA-PCR turn-around time disaggregated by county (n = 2879).**

| Transport Time: Between Collection at Health Facility and specimen received at Alupe KEMRI Laboratory | N | Mean (95% CI) (days) | p-value |
|---|---|---|---|
| Bungoma County | 600 | 17.09 (15.81; 18.36) | **<0.001** |
| Busia County | 340 | 18.99 (17.68; 20.30) | |
| Kakamega County | 1208 | 17.02(16.18; 17.86) | |
| Vihiga County | 558 | 13.01(12.14; 13.89) | |
| Total | 2706 | 16.46 (15.92; 16.99) | |
| Missing values | 173 | | |
| **Testing Time: Between specimen Received at and Tested at Alupe KEMRI Laboratory** | | | |
| Bungoma County | 608 | 17.63 (16.33; 18.93) | 0.085 |
| Busia County | 350 | 15.74 (14.38; 17.10) | |
| Kakamega County | 1237 | 17.16 (16.29; 18.04) | |
| Vihiga County | 567 | 15.79 (14.59; 17.00) | |
| Total | 2762 | 16.80 (16.23; 17.38) | |
| Missing values | 117 | | |
| **Dispatch Time: Between specimen received at and results dispatched from Alupe KEMRI Laboratory** | | | |
| Bungoma County | 608 | 25.46 (24.01; 26.92) | **0.002** |
| Busia County | 342 | 24.22 (22.32; 26.11) | |
| Kakamega County | 1239 | 27.28 (25.70; 28.86) | |
| Vihiga County | 567 | 22.89 (21.32; 24.46) | |
| Total | 2756 | 25.60 (24.72; 26.47) | |
| Missing values | 123 | | |
| **Global TAT: Between specimen collected at Health Facility and results received from Alupe KEMRI Laboratory** | | | |
| Bungoma County | 61 | 38.15 (33.81; 42.49) | **0.001** |
| Busia County | 33 | 47.76 (40.04; 55.47) | |
| Kakamega County | 326 | 51.70 (46.55; 56.86) | |
| Vihiga County | 80 | 33.66 (30.27; 37.06) | |
| Total | 500 | 46.90 (43.37; 50.43) | |

*(Continued)*

**Table 2.** (Continued)

| Transport Time: Between Collection at Health Facility and specimen received at Alupe KEMRI Laboratory | N | Mean (95% CI) (days) | *p-value* |
|---|---|---|---|
| Missing values | 2379 | | |

This table shows the descriptive characteristics of four key turnaround time variables that were abstracted from the registers and shows the mean, 95% confidence interval, and the p-values. This was done using one-way ANOVA at 5% level of significance. Missing values not included in the statistical analysis.

## B) Key informant interview results

Overall nine (9) key informant interviews were conducted. A total of five (5) health facilities laboratory-in charges, one (1) from the county laboratory in-charge and three (3) sub-county laboratory in-charges participated in the key informant interviews. Three (3) of them were females and six (6) of them males. The mean years in laboratory service were 6 years at the time of data collection.

The key informant interviews conducted revealed batching and hubbing to be common in Bungoma, Kakamega and Busia Counties as opposed to Vihiga County that tended to send the specimens directly to Alupe KEMRI Reference Laboratory through the G4S courier services that were contracted by the USAID APHIAPlus Project. In addition, most of the peripheral and remote health facilities tended to have lower caseload and would take up to three weeks to get two (2) DBS specimens from HIV exposed infants and this significantly delayed the submission of the specimens to the nearest referral hospitals who then submit the specimens to Alupe KEMRI reference laboratory through the G4S courier services.

A KII 1 in Kakamega County noted. . .". . .*we have to batch our specimens to at least 2 specimens before we submit to Health Facility 1 otherwise we would not be entitled to transport reimbursement by APHIAPlus Project*. . .."APHIAPlus project would not consider reimbursing MOH staffs if they carried only one specimens to the Health Facility 1 since it was deemed not cost-effective and they rather encouraged staff to accumulate to at least 2 or more specimens for them to be eligible for transport reimbursement.

A KII with KII 4 noted that hubbing was a key practice and..". . .*we advise facilities to bring their DBS specimens to Health Facility 2 so that we can verify that the specimens are of good quality and standards before submitting to Alupe KEMRI Reference Laboratory via G4S courier services to reduce sample rejection rates which are a problem*. . ."

KII with KII 5 noted that ". . .*pooling of samples by far away facilities is a challenge*. . .". This sentiment was shared by KII 6 who noted that health facilities that prefer hubbing delays samples to be shipped to Alupe KEMRI Reference Laboratory. KII 5 revealed that hubbing is a common phenomenon that is employed to help ensure poor quality DBS specimens are not submitted to Alupe KEMRI Reference Laboratory and but would then increase the overall TAT. However, KII 8 noted that they send "*specimens directly to Alupe KEMRI Laboratory via G4S while smaller health facilities they would ride on Vihiga County CD4 laboratory networking and submit to Health Facility 4 which is a hub site for Vihiga County*". . .

## Discussions

HIV-exposed female infants represented slightly more than half of the EID specimens in the Ministry of Health EID register. This compares favourably with a cross-sectional study from routinely collected HIV-infected infants with a positive PCR test from Government of Kenya health facilities offering PMTCT from 2016 to 2018 that showed 52.6% of HIV- exposed

infants were females [30]. However, it contradicts study done in Myanmar that reveals majority of HIV-exposed infants were males at 52% [31].

The median age of infants was 2.0 (IQR: 1.5–6.0) months. This compares favorably with a cross-sectional study from routinely collected HIV-infected infants with a positive DNA-PCR test from Government of Kenya health facilities offering PMTCT from 2016 to 2018 that revealed a median age of 3 months [30].

The primary goal of early infant diagnosis is to identify the HIV-infected child early prior to the development of clinical disease during the first months of life in order to optimize survival [27, 32, 33]. Faster TAT is universally seen as desirable. It is believed that the timelier the testing is performed, the more efficient and effective the treatment will be. TAT is the total time between specimen collection, submission, processing and dispatch of the results for patient use.

Across Africa, implementation of early infant diagnosis has been met with challenges, one of which is the long TAT of the DNA-PCR for dry blood spots results that leads to delay in initiating ART to infants and children under 18 months. This study showed a long mean TAT from specimen collection to results being received at the health facility of more than six (6) weeks. This compares with a similar study done at Kapsabet District Hospital, Nandi County in Kenya, Tanzania and in Côte d'Ivoire that showed the average number of weeks from sample collection to return of the infant DNA-PCR result as about four (4) weeks and more [34–36]. A study in Swaziland also revealed the mean TAT from test to result pick-up was longer at about nine (9) weeks [19]. Many of the sub-Saharan African countries implementing EID have a high TAT exceeding four (4) weeks [37–42]. Studies done in Myanmar and India revealed long TAT of seven (7) weeks or more in Myanmar [31] and between 29 and 53 days over the four (4) years in India [43]. However, this study contrast with study that showed TAT for processing within laboratories averaged nine (9) days in Namibia, and 3.33 weeks in Uganda. However, all countries health facility registers did not systematically document the date that the result arrived back at sites and therefore the total TAT from sample collection to result arrival at site could not be measured [44]. Study in Uganda showed TAT from sample collection to result return decreased from 49 to under 14 days [45].

The mean transport time from specimen collection at the health facilities and receiving specimens at Alupe KEMRI laboratory was about two (2) weeks with Busia County that is co-located with Alupe KEMRI Reference Laboratory having the longest transport time compared to Vihiga, Bungoma and Kakamaga counties. This compares with median time between sample collection and arrival at the central laboratory in Lusaka which was 17 days [42]. However, this contrasts with mean sample TAT from collection at site to laboratory of 1.38 days in Namibia, 5.25 days in Cambodia, and 12.6 days in Uganda over the life of the program [44]. It also contrasts with a national retrospective study done in India that showed transport time was significantly higher for states without reference laboratories (42 days) than those with reference laboratories (27 days) [43]. Busia county registered the longest transport time despite being co-located with Alupe KEMRI Reference Laboratory was due to the fact that health facilities in Busia County that submit specimens to Alupe KEMRI Reference Laboratory are poorly facilitated by the county government while health facilities that are supported by USAID AMPATH project send their specimens directly to USAID AMPATH specialised Laboratory in Eldoret for specimen processing and their data were excluded from this study. AMPATH project do not strictly follow the Ministry of Health PMTCT protocol and these did not meet the inclusion criteria.

The mean testing time from receiving the specimens from the health facilities and testing the specimens was about 2 weeks. This compares with a national retrospective study done in India that showed median testing time which varied from 6 to 21 days [42]. However, it contrasts with TAT determined in Lusaka that showed the time between specimen arrival at the

central laboratory to testing was 6 days [43]. The reason for delayed specimen processing could be due to inadequate staffing, interruptions of test kits supply, incompletely and poorly filled laboratory forms that are difficult to read and record and therefore needed to be re-written before specimen processing could begin.

The mean dispatch time from receiving specimens from the health facilities and dispatching the results to the health facilities was 3 to 4 weeks. This is largely explained by the delay in specimen processing at Alupe KEMRI Reference Laboratory for reasons already alluded to earlier.

Laboratory specimen batching and hubbing is common especially in Bungoma, Kakamega and Busia Counties as compared to Vihiga County that tended to send the specimens directly to Alupe KEMRI Reference Laboratory through the G4S courier services that were contracted by the USAID APHIAPlus Project. This compares with study done in Myanmar that revealed long TAT of seven (7) weeks or more was due to specimen batching and hoarding at the hub laboratory sites [30]. Similarly, a retrospective study of national EID in India revealed that transport time was a key bottleneck contributing to the long TAT due to hoarding of samples till sufficient number of samples were collected [42]. Therefore, batching and hubbing may contribute to the long TAT in Bungoma, Kakamega and Busia counties.

Factors shown to correlate with shorter total TATs include the practice of delivering each specimen as it is collected, direct delivery route, and continuous versus batching [46]. This is in agreement with the study findings which showed health facilities in Vihiga county as opposed to other Counties were delivering specimens as they are collected (no batching) and directly submitting to Alupe KEMRI Reference Laboratory via courier services (no hubbing at the Central health facilities). The study also revealed that TAT delays were at all levels that is pre-analytical, analytical and post-analytical. This differs with study done in Australia that showed delays in TAT are most commonly pre-analytical and post-analytical [46].

A national retrospective study done in India revealed that a longer distance between the health facility and the EID testing site was associated with long TAT [30]. However, this contradicts the current study findings that show Busia County despite being co-located with Alupe KEMRI Laboratory had the longest TAT. This could be explained by the fact that health facilities sending specimens to Alupe KEMRI Reference Laboratory are not donor funded and have to use their scarce resources to collect and transport the laboratory specimens.

Longer TAT results in delayed HIV diagnosis and further delay in initiation of ART treatment among HIV infected infants. This ultimately compromises the outcome of HIV infected children given that the aim of EID diagnosis is early diagnosis and initiation of ART among these children in order to reduce the associated morbidity and mortality.

## Limitations/strengths of the study

1. Typically turn-around-time (TAT) is measured from the time of drawing of blood to the receipt of results by caregivers. Since the MOH EID register did not capture the date when the EID results were issued to the caregivers, therefore total TAT from sample collection to the time the caregiver takes to receive the results was not determined;

2. The greatest strength of this study is that it was an observational study conducted in the real world setting of Ministry of Health facilities providing PMTCT services in the four counties in Western Kenya.

## Conclusions

The TAT for early infant diagnosis of Dry blood Spot specimen was considerably high with Vihiga County having the shortest TAT due to lack of specimen batching/hubbing. Batching at the remote health facilities and hubbing at the nearest referral hospitals significantly contributed to the delayed TAT (pre-analytical delays). In addition, delayed processing of the specimens at the laboratory also contributed to the analytical delays in TAT. TAT can be reduced by minimizing specimen batching and hubbing at the health facilities and also having a quicker specimen processing at Alupe KEMRI laboratory. This potentially will result in earlier treatment initiation and better outcomes for HIV-infected infants.

## Recommendations

1. Discourage batching/hubbing of specimens to reduce the TAT;

2. The county governments in the study areas to support point of care EID testing at the health facilities;

3. Quality improvement measures should be instituted at every step of the TAT cascade right from sample collection to results being received at the health facilities;

4. Proper documentation of the date the EID results are issued to the caregivers to allow for accurate determination of the TAT up to receipts of the results by patients/guardians.

## Supporting information

**S1 Checklist. Human participants research checklist.**
(DOCX)

**S1 File. PCR database.**
(SAV)

**S1 Text. KII Guide.**
(DOCX)

**S2 Text. KII Keys.**
(DOCX)

**S3 Text. Data abstraction survey tool.**
(DOCX)

## Acknowledgments

We are deeply grateful to the Vihiga, Bungoma, Busia and Kakamega County Health Directors, the health care workers at the health facilities and staff at Alupe KEMRI Laboratory for their cooperation and support. We are indebted to the MCH/PMTCT in-charges at the health facilities in the four counties who took part in the study. We would like to specially acknowledge the Jude Mutoro, the lead research assistant for excellent supervision of the data collection and logistical support.

## Author Contributions

**Conceptualization:** Maxwell Philip Omondi.

**Data curation:** Maxwell Philip Omondi.

**Formal analysis:** Maxwell Philip Omondi.

**Funding acquisition:** Maxwell Philip Omondi.

**Investigation:** Maxwell Philip Omondi.

**Methodology:** Maxwell Philip Omondi.

**Project administration:** Maxwell Philip Omondi.

**Resources:** Maxwell Philip Omondi.

**Software:** Maxwell Philip Omondi.

**Supervision:** Maxwell Philip Omondi.

**Validation:** Maxwell Philip Omondi.

**Visualization:** Maxwell Philip Omondi.

**Writing – original draft:** Maxwell Philip Omondi.

**Writing – review & editing:** Maxwell Philip Omondi.

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
