## [Decision Letter · Decision Letter 0]

20 Feb 2024

PONE-D-23-41242The Assessment of Dry Blood Spot –Polymerase Chain Reaction (DBS-PCR) Turnaround Time (TAT) and The Associated Factors in Western Kenya: A Cross-sectional StudyPLOS ONE

Dear Dr. Omondi,

Thank you for submitting your manuscript to PLOS ONE. After careful consideration, we feel that it has merit but does not fully meet PLOS ONE’s publication criteria as it currently stands. Therefore, we invite you to submit a revised version of the manuscript that addresses the points raised during the review process.

We look forward to receiving your revised manuscript.

Kind regards,

Timothy Omara, PhD

Academic Editor

PLOS ONE

Journal Requirements:

Omondi, M.P., Ombaka, J., Mwau, M. and Ouma, C., 2016. Mother-to-child HIV transmission using single, dual and triple ARV prophylaxis regimens and their correlates in western Kenya: chart review. African Journal of Pharmacology and Therapeutics, 5(1).

In your revision ensure you cite all your sources (including your own works), and quote or rephrase any duplicated text outside the methods section. Further consideration is dependent on these concerns being addressed.

3. We are unable to open your Supporting Information file PCR database.sav. Please kindly revise as necessary and re-upload.

Reviewers' comments:

Reviewer's Responses to Questions

**Comments to the Author**

1. Is the manuscript technically sound, and do the data support the conclusions?

Reviewer #1: Yes

Reviewer #2: Partly

2. Has the statistical analysis been performed appropriately and rigorously? 

Reviewer #1: Yes

Reviewer #2: No

3. Have the authors made all data underlying the findings in their manuscript fully available?

Reviewer #1: Yes

Reviewer #2: No

4. Is the manuscript presented in an intelligible fashion and written in standard English?

Reviewer #1: Yes

Reviewer #2: Yes

5. Review Comments to the Author

Reviewer #1: Thank you for the opportunity to review this manuscript that reports important data. The data is from 10 yrs ago and so would be better reported if it included the current context of EID in Kenya. Has anything changed since then? Is there any POC EID? Have the stats improved or worsened? Many of the references are very outdated and if used, as they were contemporaneous at the time, should also include more up to date references as discussed specifically below.

POC EID has been accepted and then recommended by the WHO in 2017 and 2021 https://www.who.int/publications/i/item/WHO-HIV-2017.16 , https://www.who.int/publications/i/item/9789240022232 so a little more on this methodology and it’s benefits should be discussed rather than one sentence as a recommendation.

This manuscript does not have line numbering as requested by the submission guidelines which makes it difficult to review.

Abstract

The statement “AIDS is beginning to reverse the decades of steady progress….” is no longer true? The child mortality is decreasing in Africa again as PMTCT has been so successful. However infants and children are still dying of HIV which is unacceptable as it treatable and preventable.

Background

Page 3

The first 2 paragraphs has a reference from 2008 and 2007 – this should be updated even if the statistics are similar, as we have to see the relevance of this data for today.

3rd paragraph – remove ‘shortly’ after delivery as it can occur anytime during breastfeeding which can be for years.

Page 4

If you say in 2009 then the last line of the first paragraph should say ‘was beginning to reverse’ – that is historical.

Paragraph 2 first sentence needs updated statistics or report that it was in 2008.

Sentence with reference 7 needs updating as POC EID is now recommended.

Last sentence in the second paragraph is incorrect as there are now many recommended POC technologies.

Paragraph 3 about DBS testing should give dates as is historical.

Page 5

Full paragraph 1 – reference 11 could be updated although statements are true today.

Paragraph 2 – Give the date that this was written 1989 as this is really historical – consider rephrasing - since 1989 it was shown that PRC…..

Paragraph 3 – ‘now expanded’ – when did this happen? And does ‘currently’ refer to 2024?

Page 6 –

Top of page ‘In May 2006’ – has this EID algorithm been updated?

First main paragraph – This is written in the present tense – does this process still happen like this – is there still a CD4 laboratory networking program? Or write it in the past tense.

2nd paragraph – Reference 12 is from 1989! – the 2010 WHO recommended 4 weeks for the EID results to get back to the caregiver https://www.who.int/publications/i/item/9789241599801

Page 7

‘many patients never receive their results’ – this also means babies may be undiagnosed with increased mobidity and mortality – and the test is a wasted cost. Receiving the result is not the end point.

‘One challenge is that the type of laboratory…….’ This is only if POC PCR technologies are not used. This should be differentiated.

2nd paragraph – as this was done long ago I suggest putting in the past tense – ‘associated factors had….’ And ‘This study sought to….’

‘The results would also guide….’ – did this happen or is the situation the same currently?

Materials and methods

Sample size – where is a) and there is a b)

Page 8

It would be good to upload the data collection tools and the questionnaires for review

Page 9

Figure 1

Age in infants is commonly given in weeks and would be clearer to interpret with the main spike at 6 weeks as expected.

Page 10

Figure 2 Should Global not be in the title of the figure for clarity?

In the figure or title there needs to be clarity that the times are the mean times. Figures should be stand alone and not need text to interpret.

Discussion

Page 14

I disagree with the first sentence on this page as a negative HIV test is very important for a mother to receive to know her baby is not infected. Please reference or consider that both positive and negative results are important for medical and psychological reasons.

It is believed the timelier the rapid testing is performed. Rapid does not make sense here and can be omitted. Rapid usually refers to a POC test.

Page 16

Paragraph 1

‘The reason for delayed specimen processing could be…..’ were the lab managers who were interviewed asked about this?

Paragraph 3

Bubbing should be hubbing?

As mentioned at the beginning – due to POC EID being recommended by WHO now there should be at least a sentence explaining what it is and how it is better rather than the first mention of it being as a single line recommendation. It solves nearly all the problems of this study.

This is important data to have published and was well collected and analysed.

Reviewer #2: The Assessment of Dry Blood Spot –Polymerase Chain Reaction (DBS-PCR) Turnaround Time (TAT) and The Associated Factors in Western Kenya: A Cross-sectional Study

Title: It indicates that this is a cross-sectional study and yet from the description it is a mixed study design. I suggest leaving out the design from the tile or revising it to what it truly is

DBS PCR is not an assay name – the correct assay name is DNA PCR, DBS is a sample type. Consider revising the title and the entire write-up to reflect this

What is the definition of TAT in this study’s context?

Abstract

AIDS is not reversing progress made in MTCT – so much progress has been made and hence the statement is misleading.

Make us of the term HIV exposed infant to refer to the study subjects.

For uniformity, decide on whether you use infant or baby. Choose one to be used throughout the paper.

Recommendation is outdated, so much progress has been made in testing HEIs and therefore this conclusion will not help. Batching is no longer being done, unless due to other reasons but not due to lack of testing capacity. POCs for DNA PCR are also available.

Background

Outdated references have been used in the entire write-up, which does not entirely reflect the current situation for HEI lab testing. There are updated and modern policies and guidelines that need to be referenced here. This entire section needs to be revised with updated information

We have more than 10 testing labs in the country, not 4 as indicated

what specific issues contribute to the long TATs at these study sites? the author only mentions batching and hubbing but is not clear on the specifics

Inclusion criteria

How did you obtain written IC from a facility? This is not possible since IC only applies to individual/group study subjects. A facility is not a subject

Why did you need mother baby pairs yet the study was retrospective? Did the authors interact directly with study subjects?

Why did you include facilities testing HEI after 2012?

Consenting

County directors cannot provide IC on behalf of facilities, perhaps you mean a letter of approval was provided by the county directors.

Results

What does global TAT mean?

How was the KII done? Where and what tools were used? Where are the demographic and distribution results of the 9KIIs? Please include the methodology, analysis and results for the KII.

Did you look at whether delayed TAT had any impact on the outcome/result? What proportion were positives?

Discussion

Please update this entire section based on current information on HEI.

Limitation

This study was conducted in 2012 – over 10 years ago. This should be a limitation since linking it to the current trends on HEI is challenging

6. PLOS authors have the option to publish the peer review history of their article (what does this mean?). If published, this will include your full peer review and any attached files.

Reviewer #1: No

Reviewer #2: No

---

## [Author Response · Author response to Decision Letter 0]

27 Feb 2024

All comments addressed. The current EID testing in the study areas has not changed thanks to POC EID not working after the EGPAF/UNITAID funding ended. County governments were to absorb the running costs of the POC EID but they have not done so leading to reagents stockouts. DBS continues to be processed at Alupe KEMRI as it were in 2010. USAID are reluctant to support the POC EID since the agreement was that County government take over the POC EID.

I have re-uploaded the SPSS PCR database and i have been able to open it on SPSS version 27.0. Kindly let us have a google meet and i show you the database can be opened through SPSS version 27.0. https://meet.google.com/igc-dpkr-beg

---

## [Decision Letter · Decision Letter 1]

1 Apr 2024

PONE-D-23-41242R1The Assessment of Deoxyribonucleic acid - Polymerase Chain Reaction (DNA-PCR) Turnaround Time (TAT) and the associated factors in Western Kenya: Mixed Method Study DesignPLOS ONE

Dear Dr. Omondi,

Thank you for submitting your manuscript to PLOS ONE. After careful consideration, we feel that it has merit but does not fully meet PLOS ONE’s publication criteria as it currently stands. Therefore, we invite you to submit a revised version of the manuscript that addresses the points raised during the review process.

We look forward to receiving your revised manuscript.

Kind regards,

Timothy Omara, PhD

Academic Editor

PLOS ONE

Journal Requirements:

Reviewers' comments:

Reviewer's Responses to Questions

**Comments to the Author**

1. If the authors have adequately addressed your comments raised in a previous round of review and you feel that this manuscript is now acceptable for publication, you may indicate that here to bypass the “Comments to the Author” section, enter your conflict of interest statement in the “Confidential to Editor” section, and submit your "Accept" recommendation.

Reviewer #1: (No Response)

2. Is the manuscript technically sound, and do the data support the conclusions?

Reviewer #1: Yes

3. Has the statistical analysis been performed appropriately and rigorously? 

Reviewer #1: Yes

4. Have the authors made all data underlying the findings in their manuscript fully available?

Reviewer #1: Yes

5. Is the manuscript presented in an intelligible fashion and written in standard English?

Reviewer #1: Yes

6. Review Comments to the Author

Reviewer #1: Review of Revision 1

Thanks you for the thorough revision of this paper. It is much improved.

Please confirm if it was DNA-PCR as some analysers use RNA-PCR and there is also TNR-PCT (total nucleic acid using both?) If different analysers were used just use PCR testing. I suggest keeping the sample type so saying PCR testing of dry blood spots where applicable.

Line 26

Title

The title still needs Polymerase Chain Reaction written out -DNA? is insufficient. This is through the text as well. Also add Infant in the title so as to specify the testing group

Suggest –

An assessment of turnaround times of infant Deoxyribonucleic acid – Polymerase Chain Reaction testing and the associated factors in Western Kenya: a mixed methods study

Line 38

Can abbreviate DNA-PCR throughout if DNA is accurate or just use PCR and comment.

Line 41

DNA-PCR turnaround time (TAT) of dry blood spots (DBS) – important to mention the sample type and write out TAT in full as it is the first time mentioned. Then TAT abbreviation thereafter, as in line 47

Line 43

Can delete design – mixed methods study (methods should be plural throughout)

Line 55

The TAT for DNA-PCR specimens

Line 61

Suggest EID is another key word

Line 67 and Line 90 have different 2022 stats of the number of children with HIV? The sources are different UNAIDS 0-14 and Unicef 0-19 so choose which one and specify ages. Suggest UNAIDS as 10-19 are adolescents and not relevant to this publication?

Results

Line 342

Mean (singular) number of days

Discussion

Line 441

TAT definition is not needed as mentioned more clearly above.

It is disappointing to hear the POC testing has not been supported as this is the future of diagnostic laboratory medicine. We continue to build up evidence and implementation practice to turn this tide. All the best in Kenya.

7. PLOS authors have the option to publish the peer review history of their article (what does this mean?). If published, this will include your full peer review and any attached files.

Reviewer #1: No

---

## [Author Response · Author response to Decision Letter 1]

1 Apr 2024

The comments raised have been duly addressed. Thank you for the opportunity

---

## [Editor Report · Decision Letter 2]

3 Apr 2024

An assessment of turnaround times of infant Deoxyribonucleic acid – Polymerase Chain Reaction testing and the associated factors in Western Kenya: A Mixed Methods Study

PONE-D-23-41242R2

Dear Dr. Omondi,

We’re pleased to inform you that your manuscript has been judged scientifically suitable for publication and will be formally accepted for publication once it meets all outstanding technical requirements.

Kind regards,

Timothy Omara, PhD

Academic Editor

PLOS ONE